# Complete chloroplast genome sequence of *Adenophora racemosa* (Campanulaceae): Comparative analysis with congeneric species

**Kyung-Ah Kim[1,2], Kyeong-Sik Cheon[3]\***

**1** Department of Biological Sciences, Kangwon National University, Chuncheon, South Korea,
**2** Environmental Research Institute, Kangwon National University, Chuncheon, South Korea, **3** Department of Biological Science, Sangji University, Wonju, South Korea

\* cheonks@sangji.ac.kr

**Data Availability Statement:** All relevant data are within the manuscript and its Supporting Information files.

## Abstract

*Adenophora racemosa*, belonging to the Campanulaceae, is an important species because it is endemic to Korea. The goal of this study was to assemble and annotate the chloroplast genome of *A. racemosa* and compare it with published chloroplast genomes of congeneric species. The chloroplast genome was reconstructed using *de novo* assembly of paired-end reads generated by the Illumina MiSeq platform. The chloroplast genome size of *A. racemosa* was 169,344 bp. In total, 112 unique genes (78 protein-coding genes, 30 tRNAs, and 4 rRNAs) were identified. A Maximum likelihood (ML) tree based on 76 protein-coding genes divided the five *Adenophora* species into two clades, showing that *A. racemosa* is more closely related to *Adenophora stricta* than to *Adenophora divaricata*. The gene order and contents of the LSC region of *A. racemosa* were identical to those of *A. divaricata* and *A. stricta*, but the structure of the SSC and IRs was unique due to IR contraction. Nucleotide diversity (Pi) >0.05 was found in eleven regions among the three *Adenophora* species not included in sect. *Remotiflorae* and in six regions between two species (*A. racemosa* and *A. stricta*).

## Introduction

Among the angiosperms, Campanulaceae are known to have the chloroplast genomes with the most structural changes, along with Geraniaceae and Fabaceae [1–11]. Among the Campanulaceae, *Adenophora* species in particular have very different chloroplast genome structures due to many rearrangements [12,13]. Although many studies have been carried out on the genus *Adenophora*, its accurate phylogenetic relationships and taxonomic position are not clear [12–19]. Therefore, it is expected that the difference in chloroplast genome structure among *Adenophora* species may be used as important information to solve the phylogenetic relationships and taxonomic positions of various species that are currently unclear.

The genus *Adenophora*, which belongs to Campanulaceae, is a perennial herbaceous plant genus with ca. 50–100 species that are distributed in temperate regions in Eurasia [12,13]. This genus is commonly called "Adenophora Radix" and is an important plant resource used as an herbal medicine [20,21].

**Funding:** This research was supported by Basic Science Research Program through the National Research Foundation of Korea (NRF) funded by the Ministry of Education (2017R1A6A3A11029236).

**Competing interests:** The authors have declared that no competing interests exist.

Among *Adenophora* species, *Adenophora racemosa* J. Lee & S. Lee, discussed in this study, is endemic to Korea and was first described by Lee and Lee [22] after collection from Mt. Odae National Park in Korea. This species is considered closely related to *Adenophora divaricata* Franch. & Sav., *Adenophora tyosenensis* Nakai ex T.H. Chung and *Adenophora pulcher* Kitam. owing to morphological characteristics such as four-leaf verticillation, regular teeth on the leaf margins, and a pale green colour of the adaxial surface of the leaf basin. However, *A. racemosa* is distinguished from *A. divaricata* in that the inflorescence is a panicle, and it is distinguished from *A. tyosenensis* and *A. pulcher* by an urceolate corolla reminiscent of that of lily of the valley (*Convallaria keiskei* Miq.) [22].

In relatively recent molecular phylogenetic studies, however, the phylogenetic relationships and taxonomic position of *A. racemosa* were not clear because it exhibited unresolved paraphyly with related taxa [13–15]. Furthermore, the phylogenetic relationships and taxonomic position of many *Adenophora* species are currently ambiguous.

In this study, therefore, we reported the complete chloroplast genome sequence of *A. racemosa*, an endemic of Korea, and compared the sequence to those of four published congeneric chloroplast genomes, i.e., those from *Adenophora divaricata*, *Adenophora erecta* S.T. Lee, J.K. Lee & S.T. Kim, *Adenophora remotiflora* (Siebold & Zucc.) Miq., and *Adenophora stricta* Miq. We found that *A. racemosa* has a previously unreported unique chloroplast genome structure caused by IR contraction, important evidence supporting its recognition as an independent species. We believe that the results of this study can be used as important information for obtaining new insights into the evolutionary history of the genus *Adenophora*. Additionally, the marker information presented in this study is considered to be very useful information for further studies aiming to determine the exact phylogenetic relationships of *Adenophora* species.

## Materials and methods

### Sample collection, DNA extraction and chloroplast genome sequencing

Since *A. racemosa* is not endangered and protected species, plant materials were collected without permission. The plant material of *A. racemosa* was collected from Mt. Gaya (35° 49' 21.5" N, 128° 07' 18.3" E) in Gyeongsangnam-do Province of South Korea, and a voucher specimen (voucher no. KWNU93473) was deposited in Kangwon National University Herbarium (KWNU).

Total DNA was extracted from approximately 100 mg of fresh leaves using a DNA Plant Mini Kit (Qiagen Inc., Valencia, CA, USA). Genomic DNA was used for sequencing on the Illumina MiSeq (Illumina Inc., San Diego, CA, USA) platform.

### Assembly and genome mapping

Chloroplast genome assembly was conducted by the *de novo* assembly protocol [23] via the Phyzen bioinformatics pipeline (http://phyzen.com). The DNA of *A. divaricata* was sequenced to produce 8,361,496 raw reads with a length of 301 bp. Low-quality sequences (Phred score < 20) were trimmed using CLC Genomics Workbench (version 6.04; CLC Inc., Arhus, Denmark). After trimming, the library for *A. racemosa* included 6,991,585 reads. Then, *de novo* assembly was implemented using the CLC Genome Assembler (http://www.clcbio.com/products/clc-assembly-cell). A total of 107,248 reads were aligned and selected form chloroplast contigs using the nucmer tool in MUMmer [24]. The draft genome contigs were merged into a single contig by joining overlapping terminal sequences of each contig. Additionally, the chloroplast genome coverage was estimated using CLC Genomics Workbench (version 6.04; CLC Inc.).

The protein-coding genes, transfer RNAs (tRNAs), and ribosomal RNAs (rRNAs) in the chloroplast genome were predicted and annotated using Dual Organellar GenoMe Annotator (DOGMA) with the default parameters [25] and manually edited by comparison with the published chloroplast genome sequences of Campanulaceae. tRNAs were confirmed using tRNAscan-SE [26]. A circular chloroplast genome map was drawn using the OGDRAW program [27].

### Phylogenetic analyses

Two genes (*rpl23* and *clpP*) among the total 78 PCGs were excluded from the phylogenetic analysis data matrix, since most of these gene regions were deleted, and only a few regions existed in the chloroplast genomes of *Adenophora* species. A total of 76 protein-coding genes from 13 species (see S1 Table for accession numbers) were compiled into a single file of 83,906 bp (S2 Table) and aligned with MAFFT [28]. Twelve Campanulaceae *s. str.* species were selected as the ingroups, and one species (*Lobelia chinensis* Lour.) was chosen as the outgroup. Maximum likelihood (ML) analyses were performed using RAxML v7.4.2 with 1000 bootstrap replicates and the GTR+I model [29]. Bayesian inference (ngen = 1,000,000, samplefreq = 200, and burninfrac = 0.25) was carried out using MrBayes v3.0b3 [30], and the best substitution model (GTR+I) was determined by the Akaike information criterion (AIC) in jModelTest version 2.1.10 [31].

### Comparative analysis of genome structure

mVISTA was used to compare similarities among the five *Adenophora* species using shuffle-LAGAN mode [32]. The annotated *A. racemosa* chloroplast genome was used as a reference. Additionally, the genome structures of the five *Adenophora* species were compared using MAUVE [33].

### Nucleotide diversity and Ka/Ks ratio analysis

To assess complete nucleotide diversity (Pi) among the five *Adenophora* chloroplast genomes, the complete chloroplast genome sequences were aligned using the MAFFT [28] aligner tool and manually adjusted with BioEdit [34]. We then performed sliding window analysis to calculate the nucleotide variability (Pi) values using DnaSP 6 [35] with a window length of 600 bp and a step size of 200 bp [36]. The 75 protein-coding genes were extracted and aligned separately using MAFFT [28] to estimate the synonymous (Ks) and nonsynonymous (Ka) substitution rates. The Ka/Ks for each gene was estimated in DnaSP [35].

## Results

### Feature of the *Adenophora* chloroplast genomes

The chloroplast genome of *Adenophora racemosa* (GenBank accession no. MT012303) has been submitted to GenBank of the National Center for Biotechnology Information (NCBI). The complete chloroplast genome of *A. racemosa* is 169,344 bp in length, with an average mean coverage depth of 159-fold (S1 Fig). It exhibits a typical quadripartite architecture, with an LSC (large single copy), an SSC (small single copy) and a pair of IRs (inverted repeats) of 122,518 bp, 29,588 bp and 8619 bp, respectively (Fig 1; Table 1).

The total length of the chloroplast genomes of five *Adenophora* species, i.e., *A. racemosa* and four species analysed in a previous study (*A. divaricata*, *A. erecta*, *A. remotiflora*, and *A. stricta*), ranged from 159,759 to 176,331 bp (Table 1). The length of the LSC regions in the five chloroplast genomes anged from 105,555 to 122,518 bp, and the SSC and IR were 8648 to 29,588 bp and 8619 to 28,098 bp in length, respectively. In the chloroplast genome of *A.*

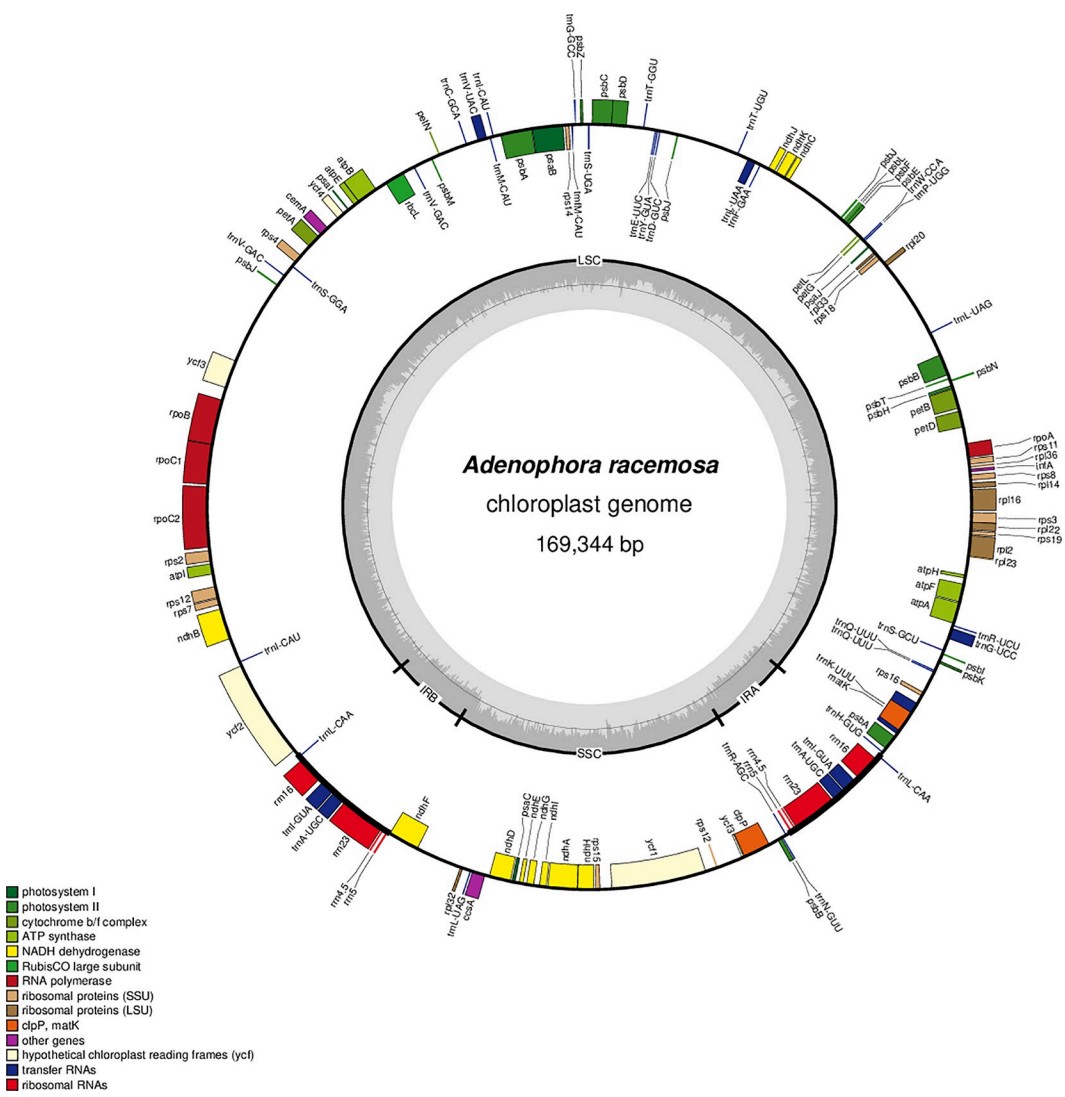

**Fig 1. Gene map of the *Adenophora racemosa* chloroplast genome.** Genes inside the circle are transcribed clockwise, and genes outside are transcribed counterclockwise. The dark grey inner circle corresponds to the GC content, and the light-grey circle corresponds to the AT content.

*racemosa*, very long sequences were inserted into two IGSs (intergenic spacers) of *psbB-rpl20* and *ψpsbJ-ycf3*, resulting in an extended LSC region (Fig 2; Table 1).

Additionally, each of the five chloroplast genomes contained 112 unique genes, including 78 protein-coding genes, 30 transfer RNAs (tRNA), and 4 ribosomal RNAs (rRNA). The G+C contents in the five chloroplast genomes ranged from 37.7 to 38.8%.

Cheon et al. [12] reported that three genes (*rpl23*, *infA*, and *clpP*) in *Adenophora* chloroplast genomes were pseudogenized, two tRNAs (*trnI-CAU* and *trnV-GAC*) and one gene (*psbJ*) had one additional copy and two additional copies, respectively, and part of three genes (*psbB*, *ycf3*, and *rrn23*) was duplicated. The *A. racemosa* chloroplast genome analysed in this study had the same characteristics. The 5' exon of the *rps12* gene in the *A. racemosa* chloroplast genome was located in the SSC region due to IR contraction, making it identical to the chloroplast genome of *A. stricta*. Meanwhile, *trnQ-UUG* in the chloroplast genome of *A. racemosa* had an additional copy in the LSC region.

**Table 1. Comparison of chloroplast genome features of five *Adenophora* species.**

| Feature | A. racemosa | A. stricta | A. divaricata | A. erecta | A. remotiflora |
|---|---|---|---|---|---|
| GenBank accession No. | MT012303 | KX462131 | KX462129 | KX462130 | KP889213 |
| Genome size | 169,344 | 159,759 | 176,331 | 173,324 | 171,724 |
| Large single copy (LSC) | 122,518 | 112,321 | 113,353 | 105,861 | 105,555 |
| Small single copy (SSC) | 29,588 | 27,238 | 8648 | 11,267 | 11,295 |
| Inverted repeat (IR) | 8619 | 10,100 | 27,165 | 28,098 | 27,437 |
| Number of unique protein-coding genes | 78 | 78 | 78 | 78 | 78 |
| Number of tRNAs | 30 | 30 | 30 | 30 | 30 |
| Number of rRNAs | 4 | 4 | 4 | 4 | 4 |
| G+C (%) | | | | | |
| Large single copy (LSC) | 36.1 | 37.1 | 37.1 | 37.5 | 37.5 |
| Small single copy (SSC) | 35.6 | 35.4 | 33.0 | 35.0 | 34.9 |
| Inverted repeat (IR) | 52.3 | 51.0 | 42.2 | 41.8 | 42.0 |
| Total genome | 37.7 | 38.5 | 38.5 | 38.7 | 38.8 |

## Phylogenetic analyses of Campanulaceae

The ML (maximum likelihood) tree formed the following two clades: platycodonoids and campanuloids. The campanuloids formed two subclades: the *Campanula s. str.* clade and *Rapunculus* clade. All nodes in the ML tree were strongly supported, with 100% BP (bootstrap) and 1.00 PP (Bayesian posterior probability) values (Fig 3).

In the *Campanula s. str.* clade, *Trachelium caeruleum* L. formed a basal branch, and *Campanula zangezura* (Lipsky) Kolak. et Serdjukova was sister to *Campanula punctata* Lam. and the *Campanula takesimana* Nakai clade. Within the *Rapunculus* clade, *Hanabusaya asiatica* (Nakai) Nakai was the earliest-diverging lineage and was sister to all other species in the clade. Additionally, five *Adenophora* species were divided into two subclades: a clade containing the sect. *Remotiflorae* species (*A. remotiflora* and *A. erecta*) and a clade containing the remaining three *Adenophora* species. Furthermore, *A. divaricata* was sister to the *A. stricta* and *A. racemosa* clade.

## The structural changes of *Adenophora* chloroplast genomes

The gene order and contents of the LSC region of *A. racemosa* were identical to those of *A. divaricata* and *A. stricta*. In the results of previous study [12], the LSC of *A. divaricata* and *A. stricta* were confirmed that inversion of two large gene blocks (*trnT-UGU-ndhC*, and *psbJ-ψpsbJ*) were occurred when compared to LSC of sect. *Remotiflorae* speices, *A. erecta* and *A. remotiflora*. Cheon et al [12] also reported that the gene order and contents of the IR and SSC in two sect. *Remotiflorae* species and *A. divaricata* were the same, but the IR of *A. stricta* was identified as being much shorter than that of other *Adenophora* species due to IR contraction. Meanwhile, the IR of *A. racemosa* was identified as the shortest among the five studied *Adenophora* species because IR contraction, including partial contraction of *psbB*, *trnN-GUU*, and *trnR-AGC*, contraction further occurred in the *A. racemosa* chloroplast genome than in the *A. stricta* chloroplast genome (Fig 4; Table 1).

## Nucleotide diversity and Ka/Ks ratio

The average nucleotide diversity (Pi) among the five *Adenophora* chloroplast genomes and all chloroplast genomes except those of the two sect. *Remotiflorae* species were estimated to be 0.087 and 0.010, respectively. Additionally, the Pi between the two chloroplast genomes of *A. racemosa* and *A. stricta*, the species with the closest phylogenetic relationship with *A. racemosa*,

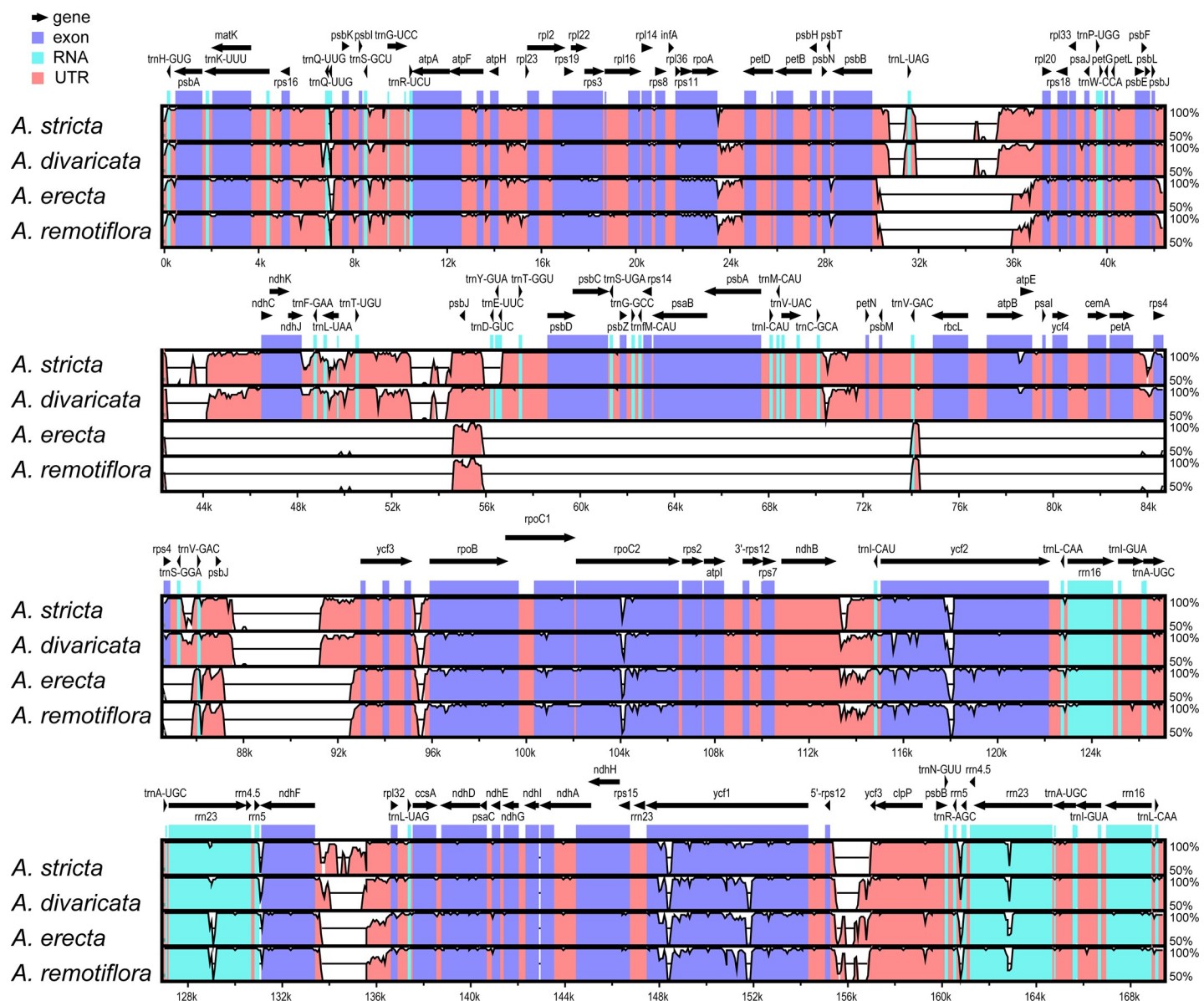

**Fig 2. Visualization of alignment of five *Adenophora* chloroplast genomes using *A. racemosa* as a reference.** The vertical scale indicates the percent identity, ranging from 50% to 100%. Coding regions, RNAs, and non-coding regions are marked in purple, sky blue, and red, respectively.

was estimated to be 0.009, ranging from 0 to 0.383 (Fig 5). In the five chloroplast genomes, seven regions (*rpoA-petD*, *psbB-rpl20*, *ycf3-ropB*, *ndhD-trnI*, *ndhF-rpl32*, and two *ycf1* regions) showed high values of Pi (> 0.05). In the results for the groups of three species and two species, 11 (*rpoA-petD*, *trnL-rpl20*, *psbJ-ndhC*, *trnT-psbJ*, *trnC-petN*, *psbJ-ycf3*, *ycf3-ropB*, *rpoC2*, *ndhF-rpl32*, and two *ycf1* regions) and seven regions (*trnL-rpl20*, *trnT-psbJ*, *trnC-petN*, *psbJ-ycf3*, *ndhF-rpl32*, and *ycf1*) showed a high value of Pi (> 0.05), respectively.

The Ka (non-synonymous)/Ks (synonymous) ratio was calculated for the 75 protein-coding genes of three *Adenophora* species, namely, *A. divaricata*, *A. stricta*, and *A. racemosa* (Fig 6; S3 Table). Comparison between *A. divaricata* and *A. stricta* revealed high values of 1 or more in seven gene regions (*matK*, *rpoB*, *rpoC1*, *rpoC2*, *ycf2*, *ndhF*, and *ycf1*), and that between *A.*

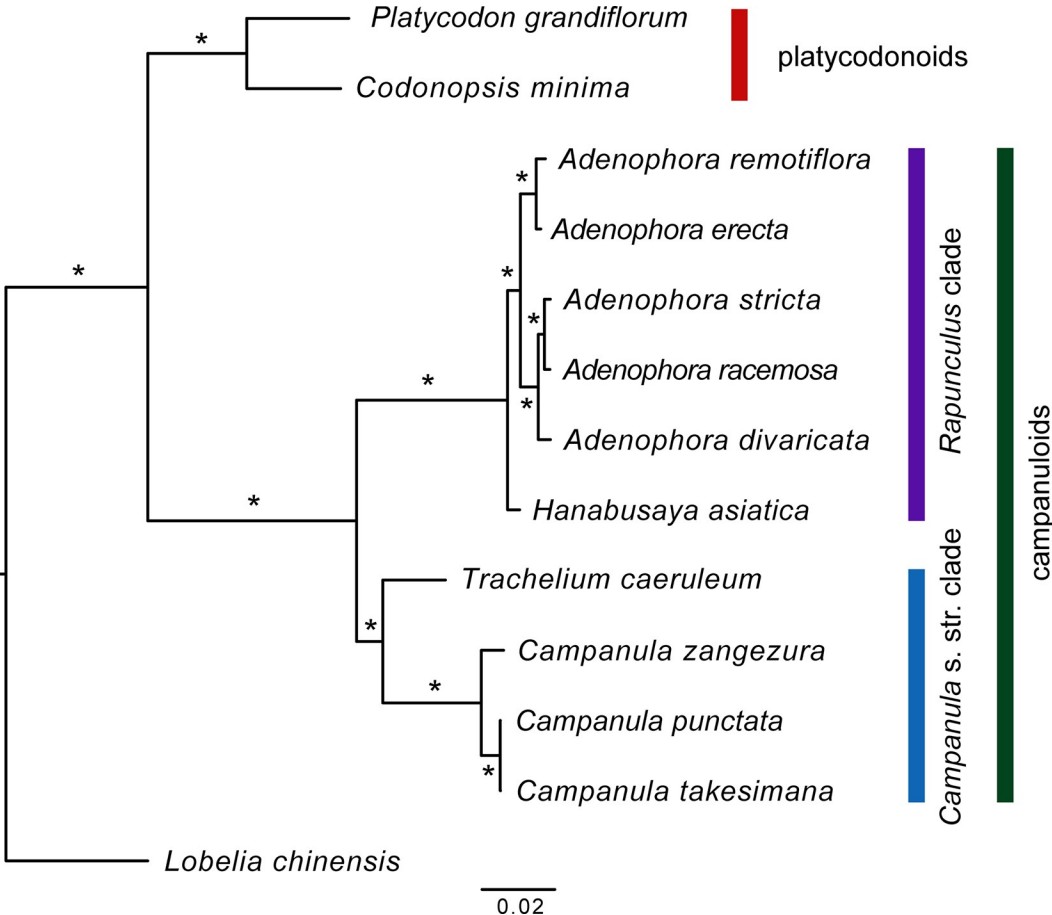

**Fig 3. The ML tree based on 76 protein coding genes from 13 chloroplast genomes.** The 100% bootstrap (BP) value and 1.00 Posterior probability (PP) value are marked with *.

*divaricata* and *A. racemosa* showed that 5 gene regions (*matK*, *rpoB*, *rpoC1*, *rpoC2*, and *ycf1*) had a value of 1 or more. Furthermore, only one region showed a high value of more than 1 between *A. stricta* and *A. racemosa*, which showed the closest phylogenetic relationship.

## Discussion

### Chloroplast genome organization in *Adenophora*

The lengths of the LSC of *A. divaricata*, *A. stricta*, and *A. racemosa* were longer than those of the two sect. *Remotiflorae* species (*A. erecta* and *A. remotiflora*). Additionally, *A. racemosa* had the longest LSC among the five *Adenophora* species. The difference in the lengths of LSC regions between sect. *Remotiflora* and the remaining three species is judged to be due to sequence mutations of the inversion end point of two large gene blocks. Also, we confirmed that the difference lengths of IRs and SSC regions among the three *Adenophora* species except two sect. *Remotiflorae* species were attributed to IR contraction (Fig 4).

Adenophora species are known to be difficult to distinguish because of their overlapping morphological characters [13]. In particular, *A. racemosa*, discussed in this study, has morphological characteristics that are very similar to those of *A. divaricata*, which makes it very difficult to distinguish the two species. Therefore, the difference in chloroplast genome structure

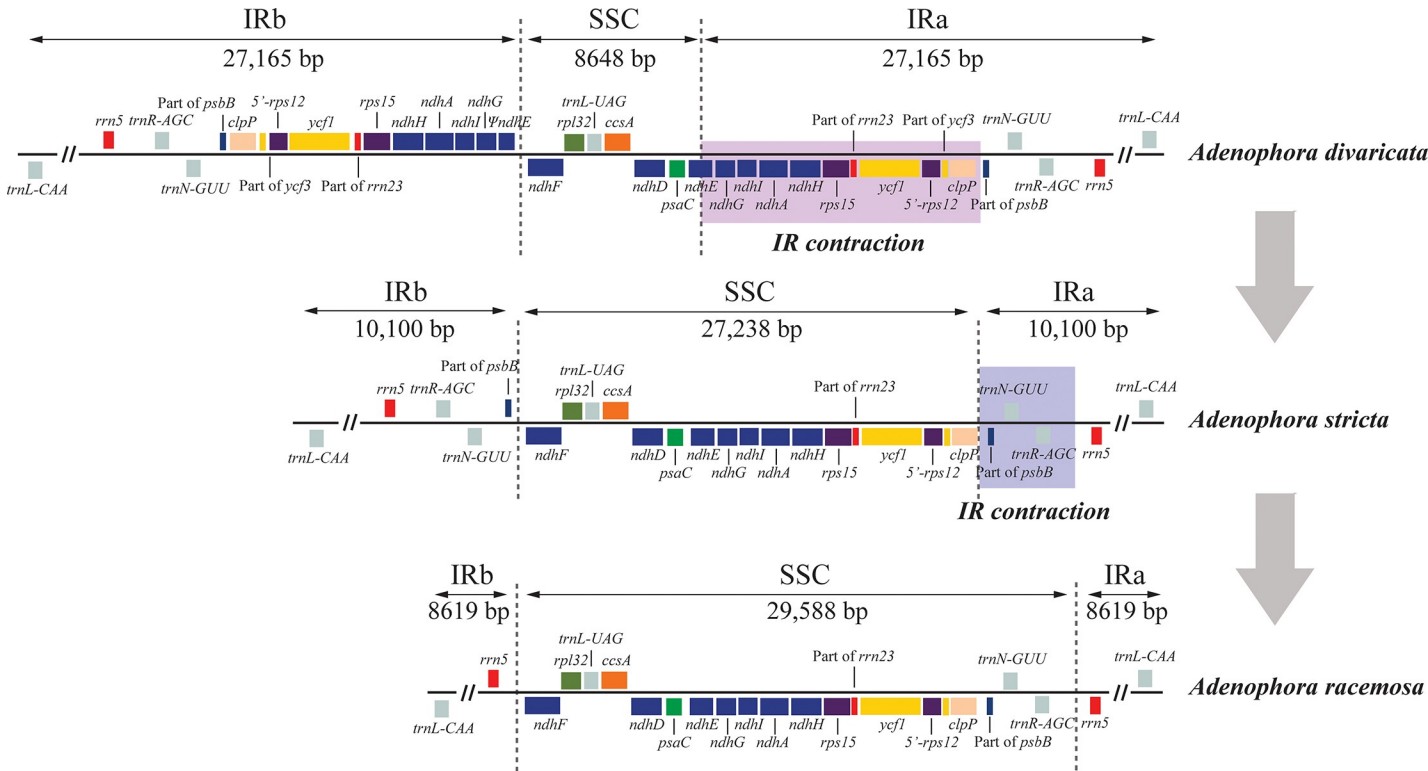

**Fig 4. IR contraction in the *Adenophora racemosa* chloroplast genome.**

between the two species identified in this study is considered to be very useful information for distinguishing between the two species.

## Suggestions for classification system of genus *Adenophora*

The ML tree in this study showed that *Adenophora* forms a monophyletic clade divided into two subclades, one containing the two sect. *Remotiflorae* species and another containing the remaining three species. In the clade containing the remaining three species, *A. racemosa* has a closer relationship with *A. stricta* than with *A. divaricata*. We think that these relationships have important implications because they are different from the relationships in the recent classification system.

The classification system of *Adenophora* has been established by many studies [37–44], and the species in this genus are divided into sections mainly by leaf arrangement and disk shape. Among the five *Adenophora* species discussed in this study, accordingly, it is common to treat *A. erecta* and *A. remotiflora* as belonging to sect. *Remotiflorae*, *A. divaricata* and *A. racemosa* as belonging to sect. *Platyphyllae*, and *A. stricta* as belonging to sect. *Gmelinianae*. However, the two species belonging to sect. *Platyphyllae* exhibited paraphyly, and these phylogenetic relationships were different from the relationships in the current classification system. Of course, this study was carried out with only a few taxa, which makes it difficult to discuss the complete phylogenetic relationships of *Adenophora*. However, paraphyletic relationships have been confirmed in this study, and we think that in-depth studies are necessary to delimit the sections of *Adenophora*, except sect. *Remotiflorae*.

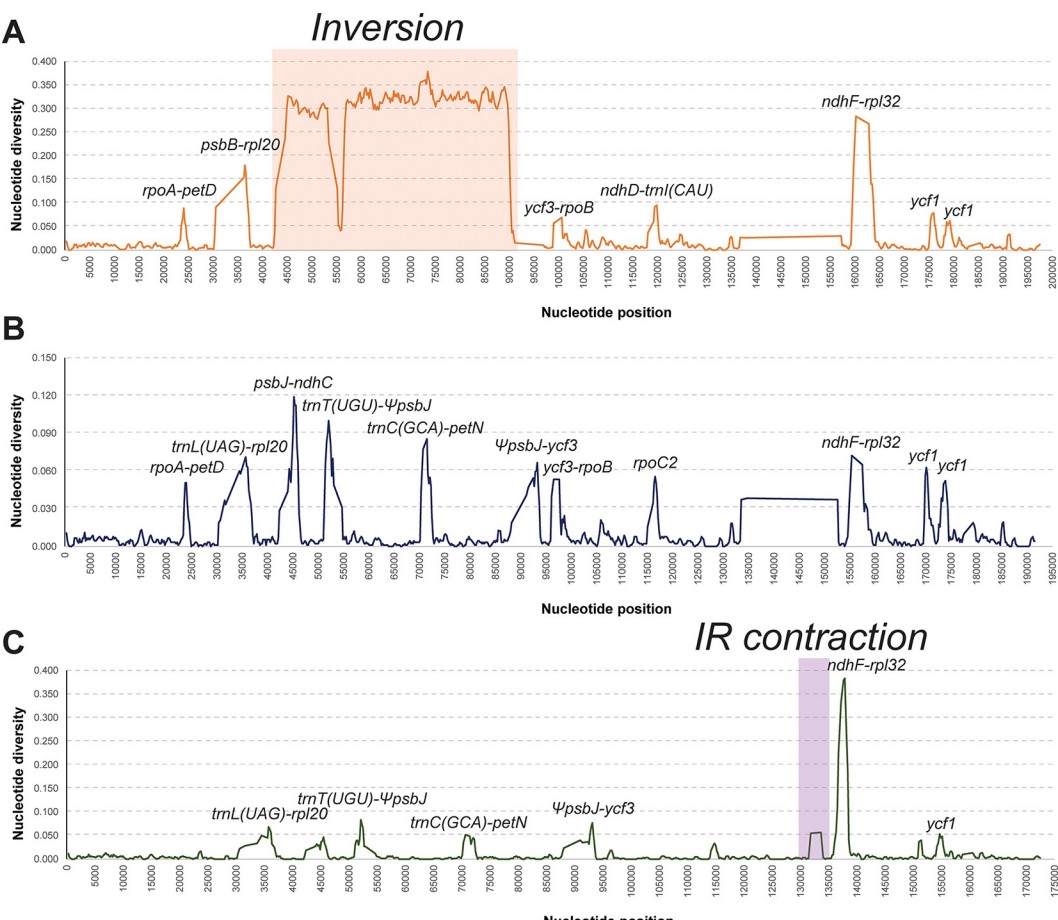

**Fig 5. Sliding window analysis of *Adenophora* chloroplast genomes.** A; Pi values of five *Adenophora* species, B; Pi values of three *Adenophora* species, excluding the two sect. *Remotiflorae* species, C; Pi values of *A. stricta* and *A. racemosa*.

### Evolution of protein-coding genes in *Adenophora* species

The Ka/Ks ratio may indicate which selection pressure is acting on a particular PCGs. Ka/Ks > 1 and Ka/Ks < 1 indicate that the gene is affected by positive selection and negative selection, respectively, and a value of 0 indicates neutral selection [36,45].

The Ka/Ks ratio of *Adenophora* species was calculated for the first time in this study. As a result, between *A. divaricata* and *A. stricta*, there were two more positively selected genes (*ycf2* and *ndhF*) than between *A. divaricata* and *A. racemosa*. Additionally, between *A. racemosa* and *A. stricta*, 62 and 12 genes were calculated to be under neutral selection and negative selection, respectively, and only 1 gene (*ycf1*) was identified as being under positive selection (Fig 4; S3 Table).

In the Caesalpinioideae of Leguminosae, known as one of the groups with the most structural changes in the chloroplast genome, four genes (*ndhD*, *ycf1*, *infA* and *rpl23*) and three genes (*psbH*, *clpP*, and *rps16*) were identified as being under positive selection [36,46], respectively. In the Convolvulaceae and Araceae, three genes (*accD*, *cemA*, and *ycf2*) and only one gene (*rps12*) were positively selected, respectively. Moreover, *ycf1* was identified as the gene with the most accelerated mutation rates among the species in this study, and *ycf1* was found to have the highest sequence mutation rates among the protein-coding genes in a previous study including sect. *Remotiflorae* species [12].

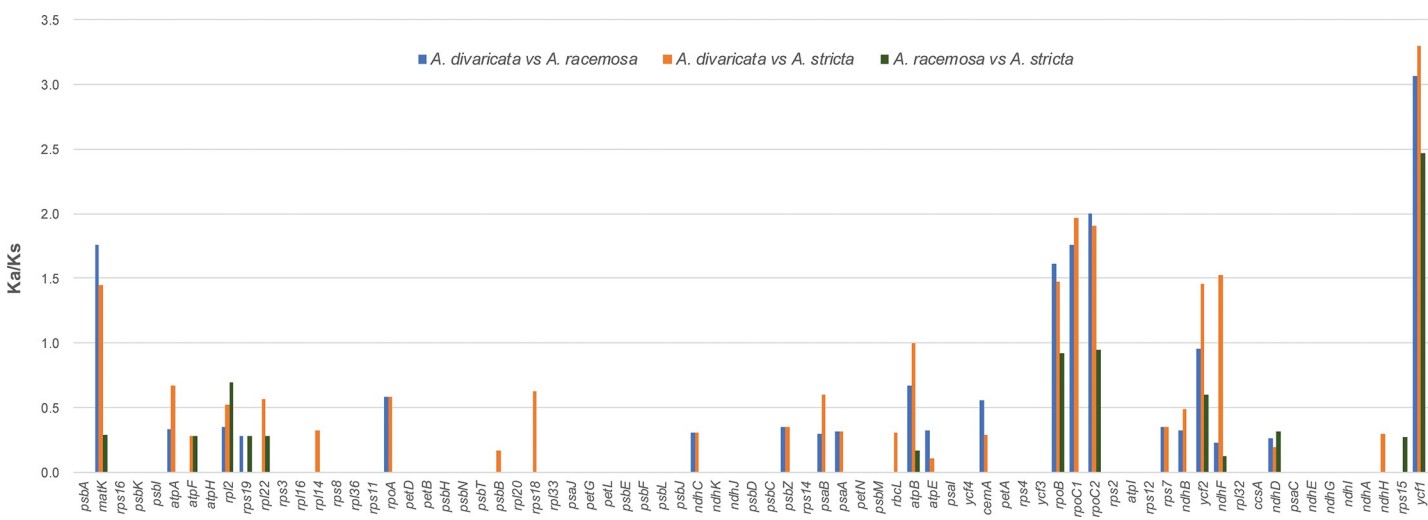

**Fig 6. The Ka/Ks ratio of *Adenophora* chloroplast genomes for individual genes.**

## Useful molecular marker information for *Adenophora* phylogenetics

We think that marker information that can best describe the phylogenetic tendencies of the remaining sections (except sect. *Remotiflorae*) is most needed at this point. In a previous study [13], because sect. *Remotiflorae* formed a monophyletic group, there was no issue in classifying it as a section.

The results of this study using the sliding window method among the three *Adenophora* species (Fig 3B) showed that the nucleotide diversity in eleven regions, including three gene regions and eight IGS (intergenic spacer) regions, had high calculated values (> 0.05). We think that six regions (Fig 3C), namely, *trnL-rpl20*, *trnT-psbJ*, *trnC-petN*, *psbJ-ycf3*, *ndhF-rpl32*, and *ycf1*, among the eleven regions have particularly high phylogenetic resolution because their nucleotide diversity values were high in two species that showed a close phylogenetic relationship in the ML tree (Fig 2).

## Conclusion

In this study, we assembled the chloroplast genome of *A. racemosa*, which had a total length of 169,344 bp. The IR of *A. racemosa* was identified as the shortest among the *Adenophora* species because of IR contraction. *A. racemosa* is not easy to distinguish because its morphological characteristics are very similar to those of *A. divaricata*. Therefore, the different structures of the chloroplast genomes are considered to be very useful information for distinguishing between the two species. The ML tree results showed that *A. racemosa* is more closely related to *A. stricta* than to *A. divaricata*, indicating a clear problem with the current classification system for *Adenophora*. Therefore, we think that further in-depth phylogenetic studies of *Adenophora* are needed, and the molecular marker information presented in this study is expected to be very useful for such studies.

## Supporting information

**S1 Fig. The mapped read depth of *A. racemosa* chloroplast genome.**
(TIFF)

**S1 Table. The GenBank accession numbers of all the 13 chloroplast genomes used for phylogenetic analysis.**
(DOCX)

**S2 Table. The length and aligned length of each gene used for phylogenetic analysis.**
(XLSX)

**S3 Table. Ka/Ks ratio of three *Adenophora* species, *A. divaricata*, *A, stricta*, and *A. racemosa*.**
(XLSX)

## Author Contributions

**Conceptualization:** Kyeong-Sik Cheon.

**Data curation:** Kyung-Ah Kim, Kyeong-Sik Cheon.

**Formal analysis:** Kyeong-Sik Cheon.

**Funding acquisition:** Kyung-Ah Kim.

**Investigation:** Kyung-Ah Kim, Kyeong-Sik Cheon.

**Project administration:** Kyung-Ah Kim.

**Resources:** Kyung-Ah Kim, Kyeong-Sik Cheon.

**Software:** Kyeong-Sik Cheon.

**Supervision:** Kyeong-Sik Cheon.

**Visualization:** Kyung-Ah Kim, Kyeong-Sik Cheon.

**Writing – original draft:** Kyung-Ah Kim, Kyeong-Sik Cheon.

**Writing – review & editing:** Kyung-Ah Kim, Kyeong-Sik Cheon.

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
