## [Decision Letter · Decision Letter 0]

3 Feb 2021

PONE-D-21-00944

Complete chloroplast genome sequence of Adenophora racemosa (Campanulaceae): comparative analysis with congeneric species

PLOS ONE

Dear Dr.Cheon,

Thank you for submitting your manuscript to PLOS ONE. After careful consideration, we feel that it has merit but does not fully meet PLOS ONE’s publication criteria as it currently stands. Therefore, we invite you to submit a revised version of the manuscript that addresses the points raised during the review process.

We look forward to receiving your revised manuscript.

Kind regards,

Tzen-Yuh Chiang

Academic Editor

PLOS ONE

Reviewers' comments:

Reviewer's Responses to Questions

**Comments to the Author**

1. Is the manuscript technically sound, and do the data support the conclusions?

Reviewer #1: Partly

Reviewer #2: Yes

2. Has the statistical analysis been performed appropriately and rigorously? 

Reviewer #1: No

Reviewer #2: Yes

3. Have the authors made all data underlying the findings in their manuscript fully available?

Reviewer #1: Yes

Reviewer #2: No

4. Is the manuscript presented in an intelligible fashion and written in standard English?

Reviewer #1: Yes

Reviewer #2: No

5. Review Comments to the Author

Reviewer #1: Dr. Cheon et al’s MS entitled “Complete chloroplast genome sequence of Adenophora racemosa (Campanulaceae): comparative analysis with congeneric species” was to assemble and annotate the chloroplast genome sequence of Adenophora racemosa, an endemic of Korea, and compare the sequence with four published chloroplast genomes of congeneric species. The MS is basically written clearly. The results are clearly presented and the discussions are basically reasonable and sound, but there are some issues need to be clarified. In addition, I do not see any significant novelties of the MS in the data analysis approach and conclusion found in this MS.

1. The genus Adenophora contains many species. Why did you select only five species from this MS for analysis? What are the selection criteria?

2. Please also use the regions with high values of Pi (> 0.05) for phylogenetic analysis, and compare with the results of the 76 protein-coding genes.

3. L21-L22: “genome” should be changed to “chloroplast genome”.

4. L22-L33: “Adenophora racemosa” should be changed to “A. racemosa”.

5. L102： In this study, 78 protein-coding genes were identified, but why 76 protein-coding genes were used for phylogenetic analyses?

6. L118: “Adenophora” should be rendered in italic font.

7. L130: “169,344” should be changed to “169,344 bp”.

8. L132: “Small single copy” should be changed to “small single copy”.

9. L165-L166: “13 Campanulaceae and one outgroup” is inaccurate, please rephrase it.

10. L198: Check if “seven regions” is correct, I’m counting “six regions”.

11. L205: Please check the “75 protein-coding genes”. The L122 in Materials and Methods described “76 protein-coding genes”.

12. L261-L264: Where possible, please rewrite these sentences to avoid high similarity to ref. [36].

13. The latitude and longitude of the sampling location is missing.

14. Line 118, italic for the genus name.

15. Line 130, change '169,344 in' to '169,344 bp in'.

16. The method in the article shows that ML and BI (Bayesian Inferce) are used to construct a phylogenetic tree (line 105-108). The result refers to the ML tree and the branch support of the two methods (BP and PP), and fig2 is ML tree. The description of the BI tree is missing here (line 159-163).

17. Line 191, change 'two Remotiflorae' to 'two sect. Remotiflorae'.

18. The comparison of the Ka/Ks ratio is only made among three species. I think we should compare all the five species in this genus, and there may be different findings, which will help understand the phylogenetic relationship between these five species.

19. Ka/Ks is the result of pairwise comparison in DnaSP. By counting and comparing the number of regions with a ratio greater than 1, it is helpful to judge the distance of relationship. But for whether a gene has undergone positive selection, a more conventional method needs to be carried out under the framework of phylogeny.

20. Line 225, change 'occurred one more time in' to 'contraction further occurred in'.

21. Line 269, the ycf1 gene should be under positive selection.

22. The formats of the cited documents are not uniform. For example, line 377 and 380, lack year information. Line 380, change '20; 3252-3255 ' to '20:3252-3255 '. And line 404, line 411, line 416, line 418, 422 Line, line 424, line 428, line 432 and so on.

Reviewer #2: Major points

1) Professional English proof-reading is recommended.

2) Re-writing of Discussion is recommended. For example, Figs 5 and 6 should be first presented and explained in Result but not in Discussion (Lines 221-226).

3) GenBank Acc. No. MT012303 is not available now.

4) It is needed to present an evidence to confirm IR contraction. For example, NGS-read mapping coverage graph on chloroplast genome or PCR validation results.

Minor points

1) Plastome, plastid genome, chloroplast genome, and cp genome; these terms are needed to be unified into a single term, such as chloroplast genome.

2) Lines 89-90: More detailed explanation is needed. Do “107,248 reads” mean the reads from chloroplast genome or chloroplast contigs? It is needed to present a reference chloroplast genome sequence used for nucmer tool-based selection.

3) Lines 269-270: “negative selection” is thought to be “positive selection”.

6. PLOS authors have the option to publish the peer review history of their article (what does this mean?). If published, this will include your full peer review and any attached files.

Reviewer #1: No

Reviewer #2: No

---

## [Author Response · Author response to Decision Letter 0]

22 Feb 2021

Response to Reviewers

We are pleased to resubmit for publication the revised version of PONE-D-21-00944 “Complete chloroplast genome sequence of Adenophora racemosa (Campanulaceae): comparative analysis with congeneric species” We appreciated the constructive criticisms of the reviewers. We have addressed each of their concerns as outlined below.

→ We checked and reflected the template style of PLOS ONE at the request of the journal.

→ We have added to the manuscript what you pointed out.

Reviewer 1.

1. The genus Adenophora contains many species. Why did you select only five species from this MS for analysis? What are the selection criteria?

→ As you mentioned, genus Adenophora consist of many species (approximately 50-100 species). However, the studies about cp genome of this genus were not conducted except for only two studies (Cheon et al. 2016, 2017), and accordingly, the number of taxa we could use for comparative analyses were very limited to 5 taxa including the one species (Adenophora recemosa) discussed in this study.

2. Please also use the regions with high values of Pi (> 0.05) for phylogenetic analysis, and compare with the results of the 76 protein-coding genes.

→ As you suggested, we performed a phylogenetic analysis using genetic markers with high Pi values. However, the tree was exactly the same as the phylogenetic tree based on 76 PCGs. This result is thought to be because the molecular markers proposed by us is a genetic regions suitable for phylogeny of genus Adenophora. Therefore, we have not added any relevant information to the manuscript. Additionally, we are preparing to conduct extensive study on the genus Adenophora using the molecular marker regions devised by us. Upon completion of this study, we promise to publish in the PLOS ONE.

3. L21-L22: “genome” should be changed to “chloroplast genome”.

→ We checked it and changed it as you suggested.

4. L22-L33: “Adenophora racemosa” should be changed to “A. racemosa”.

→ We checked it and changed it as you suggested.

5. L102： In this study, 78 protein-coding genes were identified, but why 76 protein-coding genes were used for phylogenetic analyses?

→ Because two genes (rpl23 and clpP) were existed only few gene regions in the Adenophora plastomes due to sequence deletion, we were conducted the phylogeny exclude these genes. Also, we added the information of these in the Materials and Methods section.

6. L118: “Adenophora” should be rendered in italic font.

→ We checked it and changed it as you suggested.

7. L130: “169,344” should be changed to “169,344 bp”.

→ We checked it and changed it as you suggested.

8. L132: “Small single copy” should be changed to “small single copy”.

→ We checked it and changed it as you suggested.

9. L165-L166: “13 Campanulaceae and one outgroup” is inaccurate, please rephrase it.

→ We have modified the Fig. 3. legend you pointed out as follows: The ML tree based on 76 protein coding genes from 13 plastomes.

10. L198: Check if “seven regions” is correct, I’m counting “six regions”.

→ For what you pointed out, we double-checked it, and as a result, two regions of the ycf1 gene show high value (>0.05), so a total of seven regions are correct.

11. L205: Please check the “75 protein-coding genes”. The L122 in Materials and Methods described “76 protein-coding genes”.

→ We double-check and ‘75 protein coding gene’ is correct because it was confirmed that three genes (rpl23, infA, and clpP) were pseudogenized. It is noted in the ‘Feature of the Adenophora plastomes’ in the ‘Results’ section. We also modified ‘76 protein coding genes’ to ‘75 protein coding genes’ in the Materials and Methods section.

12. L261-L264: Where possible, please rewrite these sentences to avoid high similarity to ref. [36].

→ As you pointed out, the sentences are written very similar to reference 36. Therefore, we have modified the sentences as follows: The Ka/Ks ratio may indicate which selection pressure is acting on a particular PCGs. Ka/Ks > 1 and Ka/Ks < 1 indicate that the gene is affected by positive selection and negative selection, respectively, and a value of 0 indicates neutral selection [36, 45].

13. The latitude and longitude of the sampling location is missing.

→ We added the GPS information of sampling location.

14. Line 118, italic for the genus name

→ We modified it as italic.

15. Line 130, change '169,344 in' to '169,344 bp in'.

→ We modified it.

16. The method in the article shows that ML and BI (Bayesian Inferce) are used to construct a phylogenetic tree (line 105-108). The result refers to the ML tree and the branch support of the two methods (BP and PP), and fig2 is ML tree. The description of the BI tree is missing here (line 159-163).

→ We reconstructed the phylogenetic tree using Maximum likelihood method. Also, the BP value based on bootstrap method and PP value based on Bayesian inference were performed to confirm the support values of each node in ML tree.

17. Line 191, change 'two Remotiflorae' to 'two sect. Remotiflorae'.

→ We modified it.

18. The comparison of the Ka/Ks ratio is only made among three species. I think we should compare all the five species in this genus, and there may be different findings, which will help understand the phylogenetic relationship between these five species.

→ Since the three species except two sect. Remotiflorae species are morphologically very similar, we conducted an analysis that included only three species (A. divaricate, A. stricta, and A. racemosa). We understand that including all five species as suggested by you may result in different. However, we think it is better to present more concise and intensive analysis results. Please understand with your broad generosity.

19. Ka/Ks is the result of pairwise comparison in DnaSP. By counting and comparing the number of regions with a ratio greater than 1, it is helpful to judge the distance of relationship. But for whether a gene has undergone positive selection, a more conventional method needs to be carried out under the framework of phylogeny.

→ Thank you for your valuable opinion. But we do not understand what the ‘more conventional method’ you mentioned means. If you can tell us how, we will add it in the next review.

20. Line 225, change 'occurred one more time in' to 'contraction further occurred in'.

→ We modified it.

21. Line 269, the ycf1 gene should be under positive selection.

→ We modified it.

22. The formats of the cited documents are not uniform. For example, line 377 and 380, lack year information. Line 380, change '20; 3252-3255 ' to '20:3252-3255 '. And line 404, line 411, line 416, line 418, 422 Line, line 424, line 428, line 432 and so on.

→ We modified it.

 

Reviewer 2.

Major points

1) Professional English proof-reading is recommended.

→ Manuscript has been completely revised by a professional English translation agency.

2) Re-writing of Discussion is recommended. For example, Figs 5 and 6 should be first presented and explained in Result but not in Discussion (Lines 221-226)

→ We have re-written the discussion as your suggestion.

3) GenBank Acc. No. MT012303 is not available now.

We think that GenBank accession number will be released soon. We submitted a sequin file on the chloroplast genome of Adenophora racemosa to NCBI in Feb. 2020. At the time of submission, we had requested to NCBI that genome information of A. racemosa be not released for a year because of a group doing study very similar to ours.

4) It is needed to present an evidence to confirm IR contraction. For example, NGS-read mapping coverage graph on chloroplast genome or PCR validation results.

→ We added the NGS-read mapping coverage graph as a supplementary Figure 1.

Minor points

1) Plastome, plastid genome, chloroplast genome, and cp genome; these terms are needed to be unified into a single term, such as chloroplast genome.

→ We have unified the words you pointed out as the ‘chloroplast genome’.

2) Lines 89-90: More detailed explanation is needed. Do “107,248 reads” mean the reads from chloroplast genome or chloroplast contigs? It is needed to present a reference chloroplast genome sequence used for nucmer tool-based selection.

→ ‘107,248 reads’ mean the reads from chloroplast contigs. We added this on the Materials and Methods section. 

3) Lines 269-270: “negative selection” is thought to be “positive selection”.

→ We modified it.

---

## [Decision Letter · Decision Letter 1]

5 Mar 2021

Complete chloroplast genome sequence of Adenophora racemosa (Campanulaceae): comparative analysis with congeneric species

PONE-D-21-00944R1

Dear Dr. Cheon,

We’re pleased to inform you that your manuscript has been judged scientifically suitable for publication and will be formally accepted for publication once it meets all outstanding technical requirements.

Kind regards,

Tzen-Yuh Chiang

Academic Editor

PLOS ONE

Additional Editor Comments (optional):

Reviewers' comments:

Reviewer's Responses to Questions

**Comments to the Author**

1. If the authors have adequately addressed your comments raised in a previous round of review and you feel that this manuscript is now acceptable for publication, you may indicate that here to bypass the “Comments to the Author” section, enter your conflict of interest statement in the “Confidential to Editor” section, and submit your "Accept" recommendation.

Reviewer #1: All comments have been addressed

Reviewer #2: All comments have been addressed

2. Is the manuscript technically sound, and do the data support the conclusions?

Reviewer #1: Yes

Reviewer #2: Yes

3. Has the statistical analysis been performed appropriately and rigorously? 

Reviewer #1: Yes

Reviewer #2: Yes

4. Have the authors made all data underlying the findings in their manuscript fully available?

Reviewer #1: Yes

Reviewer #2: No

5. Is the manuscript presented in an intelligible fashion and written in standard English?

Reviewer #1: Yes

Reviewer #2: Yes

6. Review Comments to the Author

Reviewer #1: I have no additional comments for this manuscript, including concerns about dual publication, research ethics, or publication ethics.

Reviewer #2: GenBank Acc. No. MT012303 is not available now. Authors should request immediately open of the sequence information to NCBI GenBank via e-mail.

7. PLOS authors have the option to publish the peer review history of their article (what does this mean?). If published, this will include your full peer review and any attached files.

Reviewer #1: No

Reviewer #2: No

---

## [Editor Report · Acceptance letter]

9 Mar 2021

PONE-D-21-00944R1 

Complete chloroplast genome sequence of *Adenophora racemosa* (Campanulaceae): comparative analysis with congeneric species 

Dear Dr. Cheon:

I'm pleased to inform you that your manuscript has been deemed suitable for publication in PLOS ONE. Congratulations! Your manuscript is now with our production department. 

Kind regards, 

on behalf of

Dr. Tzen-Yuh Chiang 

Academic Editor

PLOS ONE